# Development of New Drugs for Autoimmune Hemolytic Anemia

**DOI:** 10.3390/pharmaceutics14051035

**Published:** 2022-05-11

**Authors:** Zhengrui Xiao, Irina Murakhovskaya

**Affiliations:** Division of Hematology, Department of Hematology-Oncology, Montefiore Medical Center, Albert Einstein College of Medicine, Bronx, NY 10467, USA; zxiao@montefiore.org

**Keywords:** autoimmune hemolytic anemia, cold agglutinin disease, warm autoimmune hemolytic anemia, autoimmune hemolytic anemia treatment

## Abstract

Autoimmune hemolytic anemia (AIHA) is a rare disorder characterized by the autoantibody-mediated destruction of red blood cells, and treatments for it still remain challenging. Traditional first-line immunosuppressive therapy, which includes corticosteroids and rituximab, is associated with adverse effects as well as treatment failures, and relapses are common. Subsequent lines of therapy are associated with higher rates of toxicity, and some patients remain refractory to currently available treatments. Novel therapies have become promising for this vulnerable population. In this review, we will discuss the mechanism of action, existing data, and ongoing clinical trials of current novel therapies for AIHA, including B-cell-directed therapy, phagocytosis inhibition, plasma cell-directed therapy, and complement inhibition.

## 1. Introduction

Autoimmune hemolytic anemia (AIHA) is an immune-mediated red blood cell destruction. It is a rare condition with an estimated incidence of 0.8–3 per 10^5^/year in adults, and a prevalence of 17 per 100,000 [1].

AIHA is characterized by the antibody-mediated destruction of red blood cells with or without complement involvement, and it can be further classified as warm, cold, or mixed based on the class and thermal amplitude of the pathogenic antibodies. Warm and cold autoimmune hemolytic anemia have different pathogenesis, which results in distinct presentation, clinical course, treatment modalities, and response to therapy. In approximately 50% of cases, AIHA is associated with an underlying collagen vascular disease, lymphoproliferative disorder, infection, hematopoietic and solid tumor transplant, solid tumors, or medication, and in the other 50%, it is idiopathic [2,3,4].

Historically, treatment has been based on expert opinions or retrospective studies with a paucity of randomized and prospective clinical trials [5,6,7,8,9]. While a majority of patients with autoimmune hemolytic anemia respond to first-line therapy, many have a chronic relapsing cause that requires subsequent therapies [10,11]. Second and third-line therapies for autoimmune hemolytic anemia are associated with risk of infection and immunosuppression. In addition, hemolysis can be rapid, severe, and life-threatening. There is currently no approved therapy for wAIHA. On 4 February 2022, the FDA approved the first therapy for the treatment of adults with cold agglutinin disease (https://www.fda.gov/drugs, accessed on 1 March 2022). There remains a high unmet medical need for novel, safe, and effective therapies for AIHA. Recent advances in understanding the pathophysiology of immunologic processes that lead to red cell destruction have led to the development of novel targeted therapies focusing on the underlying mechanism of hemolytic anemia and the treatment of resistant disease. This paper will provide a brief review of the pathophysiology of warm and cold autoimmune hemolytic anemia, with a focus on current treatments and new therapies in development for this condition.

## 2. Classification

Warm autoimmune hemolytic anemia (wAIHA) accounts for 60–70% of the cases of AIHA in adults [12]. The pathogenic antibody is IgG, which binds to the red cells at core body temperature, leading to red blood cell destruction with or without complement involvement. IgG-coated red blood cells are destroyed via antibody-dependent cell-mediated cytotoxicity (ADCC), mainly through phagocytosis, in the splenic macrophages. Direct Coombs or the direct antiglobulin test (DAT), a test which evaluates the presence of immunoglobulin (Ig) and/or a complement on the red cell surface, mostly produces positive results for the presence of IgG, and in some cases C3. In 5% of cases, the DAT has negative results, and diagnosis is made on the basis of the response to empiric corticosteroid therapy [13,14]. In about 50% of the cases of wAIHA, there is evidence of complement deposition on the surface of the red blood cells [15]. Complement-coated red blood cells undergo extravascular hemolysis in the hepatic mononuclear phagocytic system via C3b receptors. Furthermore, a membrane attack complex could be formed when a large amount of complement is generated, which would lead to intravascular hemolysis. However, this process is generally limited by the complement regulatory proteins CD55 and CD59, which are on the surface of red blood cells.

Cold AIHA is seen in 20–25% of cases of AIHA. Most cases are represented by cold agglutinin disease (CAD), a low-grade lymphoproliferative disorder that is distinct from other low-grade non-Hodgkin’s lymphoma such as lymphoplasmacytic or marginal-zone lymphoma [11,16,17]. Cold autoimmune hemolytic anemia that develops in the setting of autoimmune disease, viral infection, or malignancy is classified as cold agglutinin syndrome (CAS). The pathogenic antibody is a cold agglutinin, which in most cases is of the immunoglobulin M (IgM) subtype with I antigen specificity. IgM, which is a strong complement activator, binds to red blood cells at 4–34 °C, fixes the complement, and dissociates from the red cells as they move centrally. Activation of the classical complement pathway leads to the deposition of C3b onto the red blood cell surface, with extravascular hemolysis in the liver [16] being the main mechanism of red cell destruction. DAT has positive results for C3. The completion of the complement cascade with the formation of the membrane attack complex and intravascular hemolysis can be seen as well [17,18].

Mixed AIHA is characterized by the presence of both pathogenic IgG antibody and clinically significant cold agglutinin, and it is seen in 5–10% of cases of autoimmune hemolytic anemia. Paroxysmal cold hemoglobinuria (PCH) is the least common type of AIHA, which is seen in 1–5% of cases. PCH is most commonly seen in association with viral infections in children, and it is characterized by a biphasic IgG that binds to RBCs in the cold and causes severe intravascular hemolysis at 37 °C [1]. The IgG antibodies in wAIHA are always polyclonal, even in cases seen in association with lymphoproliferative disorders. In contrast, in cold agglutinin disease, IgM is monoclonal in the majority of the cases [11]. 

The clinical severity of the disease varies from mild, compensated hemolysis to severe life-threatening anemia, the need for transfusions, and thrombotic complications [11,19,20,21,22]. Low hemoglobin at presentation is associated with the increased risk of multiple relapses and the requirement for subsequent lines of therapy. Inadequate erythropoietic response to reticulocytopenia is seen in more than half of the cases and is associated with lower hemoglobin levels. In addition, complement involvement (warm IgG + complement, mixed, and CAD) is associated with lower hemoglobin levels and a higher risk of relapse. Third-line therapy is required in 15%, 19%, 33%, and 24% of patients positive for IgG alone, IgG + C positive, mixed WAIHA, and CAD, respectively, and 8% of cases require further lines of therapy [15].

## 3. Treatment of Warm Autoimmune Hemolytic Anemia

### 3.1. Corticosteroids

Initial therapy for warm autoimmune hemolytic anemia involves corticosteroids. The mechanisms of action of corticosteroids in AIHA include suppressing the production of autoantibodies [23], and downregulation of the number of monocyte Fcγ receptor expression [24]. Prednisone at 1–1.5 mg/kg/day results in a response rate of 70–80% [10,19]. After an initial 3–4 weeks of therapy, steroids are tapered over 4–6 months, with more rapid tapering associated with a higher risk of relapse [2,4,21]. Up to 20–30% of patients have a durable remission after the initial therapy, but the rest have a chronic, relapsing course [19,25] that requires subsequent therapy [10,15], with 10–20% either not responding to corticosteroids or requiring unacceptably high doses. Failure to respond to steroid therapy after 3 weeks, relapse on steroid tapering, or the inability to taper prednisone to less than 10–15 mg/day warrant the initiation of a second line of therapy.

### 3.2. Rituximab

Rituximab is a chimeric antibody directed against the CD20 transmembrane protein that is present in almost all B cells, from the stage of commitment to B-cell lineage to differentiation into plasma cells [26,27]. Rituximab is able to induce B-cell apoptosis, which leads to B-cell depletion that lasts for 6–12 months [26,27,28,29], suppressing the production of autoantibodies, cytokine secretion, and antigen presentation [30]. Rituximab additionally has an immunomodulatory effect on both cellular and innate immunity. Rituximab therapy has a reported overall response rate (ORR) of 70–80% in a relapsed setting, with 25–75% complete responses and a median response duration of 1–2 years. Majority of patients respond within 4 weeks after the first dose [31,32,33], but responses up to 3 months can be seen. Retreatment with rituximab is effective as well [33,34,35]. Two prospective randomized clinical trials demonstrated significantly higher response rates and relapse-free survival with the addition of rituximab to the first-line corticosteroid therapy. Responses were similar with four weekly doses of rituximab at 375 mg per square meter of body surface area, or two fixed doses of 1000 mg administered at a 2-week interval [6,36]. A fixed, low dose rituximab at 100 mg administered weekly in 4 doses demonstrated comparable efficacy in a prospective multicenter Italian trial [37]. Rituximab therapy is recommended as a second-line treatment for relapsed wAIHA and should be considered in the first line in severe cases as well as in patients for whom long-term corticosteroid therapy is not desirable [4]. Complications of Rituximab therapy include infusion reactions, neutropenia, and hypogammaglobinemia, although infections are not common [38]. Hepatitis B reactivation has been reported, and pretreatment serologic screening and prophylactic therapy in carriers are recommended [4].

### 3.3. Subsequent Therapies

Third-line therapies for warm autoimmune hemolytic anemia include splenectomy and immunosuppressive therapies. Spleen plays a role in AIHA in several ways. First, the specific honeycombed structure of the spleen can bring the red blood cell into close contact with the macrophage system, which facilitates phagocytosis. Second, the spleen contains a substantial pooling of red blood cells. Finally, the spleen is a source of antibody-producing B cells [39]. Splenectomy has initial response rates of 60–70% [40,41], with a surgical mortality rate of 0.8% in recent studies [42]. Long-term cure rates are only approximately 20% [2], and risk of infections due to encapsulated bacteria is 3.3–5%, with a 50% associated mortality rate [43,44]. Conventional immunosuppressive therapies such as azathioprine, cyclosporine, and cyclophosphamide have been reported to be effective. Azathioprine is a purine analog that affects lymphocyte production by blocking purine synthesis; it additionally induces T lymphocyte apoptosis through its interaction with the small GTP-binding protein Rac1 to downregulate the antiapoptotic Bcl-xL gene expression [45]. Cyclosporine is an immunomodulatory agent that targets T cells and inhibits cytokine gene transcription via formation with cyclophilin to decrease the phosphatase activity of calcineurin [46]. Cyclophosphamide is an alkylating agent and is able to selectively suppress regulatory T cells when used at a low dose [47]. However, the data of these conventional immunosuppressants are limited to retrospective series and case reports and are potentially influenced by concurrent corticosteroid therapy [10,48]. Danazol has also been reported to produce responses in retrospective studies [49]. Recombinant erythropoietin has been successfully used in patients with refractory AIHA, particularly in the presence of reticulocytopenia [50]. 

## 4. Treatment of Cold Agglutinin Disease

Supportive measures such as cold avoidance play an important role in the management of patients with cold agglutinin disease, and a subset of patients with mild anemia or compensated hemolysis can be observed without pharmacologic therapy [4,50,51] Severe cases of life-threatening hemolysis may require plasmapheresis. Corticosteroids are not recommended as chronic therapy in CAD due to their limited efficacy and the need for high doses to maintain remission [11,50,51,52]. Splenectomy is generally not effective since the clearance of C3b-opsonized red blood cells takes place in the liver. 

Therapies directed against the pathogenic B-cell clone has been the mainstay treatment for cold agglutinin disease. Rituximab is the most commonly used therapy, with a single-agent overall response rate of ~50% (mainly partial) in prospective clinical trials and a median response duration of 1–2 years [7,11,53]. The addition of oral fludarabine to rituximab therapy is associated with high response rates (76% overall, complete response rate of 21%) and durable remissions [9], but is associated with a high rate of hematologic toxicity, infectious complications as well as late toxicities [11].

A combination of rituximab with bendamustine has been prospectively studied in CAD, with overall and complete response rates of 71% and 40%, respectively, and the deepening of responses on follow up, with CR and PR rates of 78% and 53%. The median time to a response was 1.9 months, and the estimated median response duration was more than 88 months. Toxicities were mainly hematologic, with infections seen in 11% of patients [8,11]. 

A single cycle of bortezomib-induced responses, including complete response, was demonstrated in 32% of patients with CAD in a prospective study [54]. In a small retrospective study, ibrutinib has been reported to control hemolysis, improve hemoglobin, and induce acrocyanosis in all patients with CAS/CAD, with the majority of responses being complete [55]. Sutimlimab, a humanized monoclonal antibody which selectively targets the C1s protein, rapidly controlled hemolysis, improved hemoglobin, and reduced the need for blood transfusion in patients with CAD and a history of blood transfusion has recently became the first FDA-approved therapy for the treatment of cold agglutinin disease [51].

## 5. Novel Agents

Advancement in the understanding of signal transduction in the immune system has led to the development of new classes of agents that target various aspects of effector cell function. Signaling via B-cell receptors (BCR), T-cell receptors (TCR), and Fc receptors (FcR) share conceptual similarities. Receptor engagement results in the recruitment of the Src family of kinases, which phosphorylate transmembrane proteins with cytoplasmic domains containing immunoreceptor tyrosine-based activation motifs (ITAMs) that are associated with these receptors. This leads to the recruitment and activation of the spleen tyrosine kinase (Syk) in the case of BCR or FcR, or a homologous tyrosine kinase, ZAP70, in the case of TCR, and the subsequent phosphorylation of multiple substrates, thus activating pathways that lead to B-cell and T-cell activation, proliferation, and survival as well as phagocytosis, degranulation, cytolysis, cytokine upregulation, and antigen presentation [56,57,58,59]. 

Novel approaches to the therapy of autoimmune hemolytic anemia include B-cell-directed therapies, phagocytosis inhibition, complement inhibition, and plasma cell-directed therapies.

### 5.1. B-Cell-Directed Therapies

#### 5.1.1. PI3K Inhibition 

The phosphoinositide 3-kinases (PI3Ks) were first discovered in 1987 in relation to platelet-derived growth factor stimulation and polyoma middle T antigen transformation [60]. Three classes of PI3Ks have been identified in human cells. Class 1 PI3Ks is the most characterized class in metabolism, inflammation, cell survival, motility, and cancer progression [61]. Class 1 PI3Ks consist of a regulatory subunit and a catalytic subunit. They receive the signals from activated receptor tyrosine kinases [62], G protein-coupled receptors, and RAS. Once activated, the regulatory subunit would activate the catalytic subunit, which would further phosphorylate the phosphatidylinositol (PIP)2 to PIP3, which is a second messenger that regulates downstream kinases, including AKT, 3-phosphoinositide-dependent protein kinase 1 (PDK1), and the mammalian target of rapamycin (MTOR) [63,64]. 

There are four isoforms of the class I PI3K catalytic subunits: p110α, p110β, p110γ, and p110δ. Isoforms p110γ and p110δ are selectively expressed in hematopoietic cells and play crucial roles in the functions of both innate and adaptive immune cells. PI3Kδ is essential for B-cell development, proliferation, survival, activation, chemotactic migration, antibody production, and antigen presentation. The pathogenesis of several B-cell autoimmune diseases has been linked to aberrant PI3Kδ signaling [59]. The inhibition of p110δ blocks B-cell migration, adhesion, survival, activation, and proliferation, impairs the ability of B cells to act as APCs and activate autoreactive T cells, and reduces the secretion of pathogenic autoantibodies and pro-inflammatory cytokines [65].

Idelalisib is a first-in-class, oral, small-molecule inhibitor of PI3Kδ, which is used in treating B-cell lymphoproliferative disorders [66,67]. Idelalisib has been reported to be effective in autoimmune cytopenias, including autoimmune hemolytic anemia in the setting of chronic lymphocytic leukemia (CLL). In retrospective series and case reports, idelalisib has successfully controlled CLL-associated autoimmune cytopenias in most patients [68,69] (Table 1). However, in the majority of cases, idelalisib was used in combination with rituximab and corticosteroids, confounding the role of idelalisib as an immunomodulatory agent.

Parsaclisib is a potent, highly selective, next-generation PI3K delta inhibitor with reported single-agent activity in relapsed and refractory B-cell malignancies, including follicular, marginal zone, and mantle-cell lymphomas [70]. In addition to its direct antitumor activity, parsaclisib also functions as an immunomodulator. In a preclinical study, parsaclisib was able to inhibit B-cell proliferation, reduce the percentage of regulatory T cells, delay the terminal differentiation of CD8+ T cells, and modulate immune cells [71]. Parsaclisib has shown efficacy in preclinical models of autoantibody-mediated diseases. In an animal model, parsaclisib decreased the autoantibody formation as well as improved the inflammatory environment in lupus erythematosus and Sjogren’s syndrome mice [72]. In the preliminary report of a prospective open-label Phase 2 study on the use of parsaclisib in AIHA, 7 patients (33%) achieved CR, and 14 (67%) had PR at any of the visits in weeks 6–12. Responses were durable and seen as early as Week 2. Treatment was generally well-tolerated, with the most common treatment-emergent adverse events (TEAEs) being headache, pyrexia, and diarrhea. TEAEs led to treatment discontinuation in one patient [73]. A phase 3 randomized, double-blinded clinical trial to evaluate the efficacy and safety of parsaclisib in primary warm AIHA is currently ongoing [74].

**Table 1 pharmaceutics-14-01035-t001:** B-cell-directed therapies.

Medication	Type of AIHA	Type of Study	Concurrent Disease	Regimen	Efficacy/Response Rate	Reference
Idelalisib	Mixed	Case report	CLL	175 mg bid, rituximab, steroids	Hemoglobin increment 3.6 g/dL	[68]
	Warm	Retrospective, N = 19	CLL	150 mg bid ± rituximab	ORR 95%, 71% discontinued steroids	[69]
Parsaclisib	Warm/cold/mixed	Open-label, phase 2	N/A	1 mg daily, 2.5 mg daily	CR 33%, PR 66.7%	[73]
Ibrutinib	N/A	Retrospective, N = 21	CLL	420 mg daily	Hemoglobin increment 2 g/dL	[75]
	Warm	Case report	MCL	560 mg daily	Hemoglobin increment 4.1 g/dL	[76]
	Mixed	Pilot study, N = 2	N/A	280 mg daily and 420 mg daily	CR 100%	[77]
	Cold	Retrospective, N = 10	CLL/SLL, WM	N/A	CR 90%, PR 10%	[78]
Sirolimus	Mixed	Case report	Post allo-SCT	3 mg/m^2^ on day 1 followed by 1 mg/m^2^ daily	CR	[79]
	Mixed and warm	Case series, N = 4	Solid organ transplant	N/A	Response rate 100%	[80]
	N/A	Retrospective, N = 14	N/A	1–3 mg daily	ORR 85.7%, CR 57.1%	[81]
	N/A	Prospective, N = 2	N/A	2–2.5 mg/m^2^ daily	CR 50%, PR 50%	[82]

allo = allogeneic, CLL—chronic lymphocytic leukemia, CR = complete response, MCL = mantle cell lymphoma, ORR = overall response rate, PR = partial response, SCT = stem cell transplant, SLL = small lymphocytic leukemia, WM = Waldenstrogm’s macroglobinemia.

#### 5.1.2. BTK Inhibition

Bruton’s tyrosine kinase (BTK) is a non-receptor cytoplasmic protein tyrosine kinase which contains five structural domains, including a pleckstrin homology (PH) domain at the N-terminus, followed by cysteine and the proline-rich Tec homology (TH) domain, Src homology domains SH2 and SH3, and the C-terminal kinase catalytic domain [83,84]. BTK is expressed in cells of the B-cell lineage, including marrow-derived hematopoietic stem cells, common lymphoid progenitor cells, and developing B cells as well as other cells of hematopoietic lineage, except T cells, natural killer cells, and plasma cells [85,86]; it also plays a key role in B-cell development. Since it was first described in the X-linked agammaglobulinemia, BTK has been widely characterized as a critical mediator of the BCR signaling pathway. Once activated by upstream SYK, BTK subsequently activates downstream signaling pathways, including the AKT pathway, and turns on pro-survival transcription factors, including NF-κB [87]. The activation of BTK regulates B-cell proliferation, maturation, migration, survival [88], and recruitment of innate immune cells; it also regulates Fcγ receptor (FcγR) signaling in macrophages and Fcε receptor (FcεR) signaling in mast cells and basophils, and it is involved in TLR and chemokine receptor signal transduction [89]. The aberrant activation of B cells and innate immunity plays a central role in the pathogenesis of autoimmune diseases and inflammation, and given the central role of BTK in BCR and FcR signaling pathways, BTK inhibition is a promising target for immunomodulatory therapy [90].

BTK inhibitors, including first-generation ibrutinib, second-generation acalabrutinib, and zanubrutinib are approved by the FDA in the treatment of B-cell lymphoproliferative neoplasms. In addition, due its immunomodulatory effect, ibrutinib was approved by the FDA in treating the steroid-resistant chronic graft-versus-host disease (GVHD) [91]; more studies are ongoing to use the BTK inhibitors to address other autoimmune disorders [92]. In a preclinical study, ibrutinib and acalabrutinib were able to decrease the production of autoantibody in the AIHA murine model [93], supporting the role of BTK inhibition as a therapeutic target in AIHA.

Ibrutinib is a highly potent and irreversible inhibitor of BTK that forms a covalent bond with a cysteine residue at the active site of BTK, which leads to the inhibition of BTK activity [94]. Ibrutinib also inhibits several other enzymes that contain homologous cysteine residues, which accounts for off-target effects such as rash, atrial fibrillation, diarrhea, and bleeding [95]. The use of ibrutinib in patients with CLL and autoimmune cytopenia (AIC), including AIHA, was associated with the improvement of cytopenia [75] and the ability to discontinue or de-escalate AIC therapy [80,96,97]. Ibrutinib was also reported to be effective in treating AIHA in the setting of mantle-cell lymphoma [76]. In a retrospective study of 10 patients with cold AIHA (in the setting of underlying lymphoproliferative disorders, including CLL/SLL, Waldenstrom macroglobulinemia (WM) in 9 patients) treated with ibrutinib, nine patients achieved CR, and one patient achieved a partial response (PR) after the initiation of ibrutinib [78] (Table 1). In a pilot study of two patients with primary transfusion-dependent AIHA refractory to multiple lines of therapy, ibrutinib use was associated with complete response (CR), with incomplete hemolysis recovery at week 16 in both patients [77]. Ibrutinib is currently being investigated in two open-label Phase 2 clinical trials for AIHA—the ISRAEL trial evaluating a combination of rituximab and ibrutinib in refractory CLL-associated AIHA [98], and as a single agent in refractory/relapsed primary AIHA [99].

Acalabrutinib is a second-generation, highly selective, potent, and covalent BTK inhibitor approved by the FDA for refractory/relapsed mantle-cell lymphoma and CLL [100]. Compared to first-generation ibrutinib, it has minimal off-target activity with a more favorable toxicity profile. In a phase 2 study of acalabrutinib in refractory/relapsed CLL, out of 11 patients with a history of AIHA or immune-mediated thrombocytopenia before therapy initiation, only one had recurrent cytopenia during therapy, suggesting the potential efficacy of acalabrutinib [101]. A phase 2 trial to assess the efficacy of acalabrutinib in treating AIHA in patients with CLL is currently ongoing [102].

Rilzabrutinib is a potent, oral, reversible covalent BTK inhibitor with a favorable safety profile designed for immune-mediated diseases [96]. It received an orphan drug designation for the treatment of pemphigus vulgaris in 2017 and for immune thrombocytopenia (ITP) in 2018 [97]. In a preclinical study, in addition to inhibiting BTK, rilzabrutinib inhibited the activation and inflammatory activity of B cells without causing lymphodepletion and blocked IgG and IgE autoantibody-mediated FcγR and FcεR signaling. Platelet aggregation was not affected [103]. In a phase 1/2 study of 45 heavily pretreated patients with relapsed refractory ITP, 40% achieved the primary endpoint of ≥2 consecutive platelet count increase of >50 × 10^9^/L and >20 × 10^9^/L without rescue medication. Responses were rapid and durable with acceptable adverse effects [104]. A phase 3 LUNA3 randomized placebo controlled trial in ITP is currently ongoing [105]. Rilzabrutinib is also being investigated in a phase 2, open-label clinical trial in patients with warm AIHA who failed corticosteroid therapy [106].

#### 5.1.3. mTOR Inhibition

The mammalian target of rapamycin (mTOR) forms two complexes—mTOR complex 1 (mTORC1) and mTOR complex 2 (mTORC2). mTORC1 mainly regulates cell growth and metabolism, while mTORC2 controls cell proliferation and survival. mTOR can integrate signaling from the presence of nutrients, growth factors, and intracellular signaling to regulate cell growth and proliferation [107]. mTOR is a component of the PI3K signaling pathway, operating both upstream and downstream of AKT at a key junction in the PI3K pathway [108]. PI3K generates PIP3, which activates AKT and leads to the phosphorylation of mTOR [109]. The PI3K/AKT/mTOR complex is a fundamental driver of cell growth, metabolism, and proliferation as well as B-cell differentiation, proliferation, and antibody production [110].

Sirolimus (Rapamycin) is a macrolide antibiotic with antifungal properties discovered in the 1970s, which also possesses potent immunosuppressive activity [111]. Sirolimus interacts with a family of intercellular binding proteins termed FK binding proteins (FKBP); the sirolimus:FKBP complex binds directly to mTOR and blocks its function, resulting in the arrest of the cell cycle in the G1 phase [111]. In addition to its inhibitory effect on T-cell proliferation, sirolimus inhibits the differentiation and proliferation of B cells, which contributes to its immunomodulatory effect. Sirolimus was used initially as an immunosuppressive agent in the setting of solid organ transplantation and now has multiple clinical applications in malignant and non-malignant disorders [112]. Sirolimus has been particularly effective for AIHA in the setting of both hematopoietic stem cell and solid organ transplantation in a pediatric population [79,80,113,114,115,116] (Table 1). In primary relapsed/refractory AIHA, Evans syndrome, and ITP, the use of sirolimus was associated with CR and OR rates of 46.7% and 75.6%, respectively, in a retrospective series. Most responses were seen within 2 months and peaked after 6 months of sirolimus therapy. There was a trend for AIHA patients to respond later but better and to relapse less [81]. In a prospective multicenter study of 30 patients with refractory autoimmune cytopenias, single-agent sirolimus had a significant effect on cytopenia, with a CR rate of 67% (8/12) in patients with thrombocytopenia. One out of two patients with isolated AIHA achieved CR, and one achieved PR. All twelve children with autoimmune lymphoproliferative syndrome (ALPS) achieved a durable CR [82]. A trial evaluating the combination of sirolimus with trans-retinoic acid (ATRA) for autoimmune anemia, including AIHA, Evans syndrome, and acquired pure red aplastic anemia is currently in progress [117].

### 5.2. Phagocytosis Inhibition

#### 5.2.1. Syk Inhibition

Spleen tyrosine kinase (SYK) is a non-receptor cytoplasmic tyrosine kinase that is highly expressed by all hematopoietic lineage cells, including B cells, T cells, macrophages, and platelets. It contains two SRC homology 2 (SH2) domains and a kinase domain [56]. The phosphorylation of the ITAMs creates a docking site for the tandem SH2 domains of Syk, allowing Syk to bind to and phosphorylate additional ITAM tyrosines, thus amplifying the signaling output of the receptor [118]. In addition, Syk triggers downstream processes, including calcium and protein kinase C signaling, RAS homologue (RHO)-family and Pyk2-mediated cytoskeletal rearrangement, and PI3K-mediated TEC-family and AKT signaling pathway activation [56]. SYK inhibition has the potential to target a number of cellular processes that these pathways regulate, such as phagocytosis, cytokine production, cytolysis, degranulation, proliferation, B-cell activation, proliferation, maturation, survival, and antigen presentation [57,58,119,120,121].

Fostamatinib is a potent adenosine triphosphate-competitive inhibitor of the Syk catalytic domain. The inhibition of Syk signaling via FcγR in macrophages blocks cytoskeletal rearrangement, which prevents degranulation and antigen internalization. In addition, the inhibition of Syk-dependent B-cell receptor signaling has the potential to reduce cytokine and pathogenic antibody production. In early preclinical studies, fostamatinib was also able to inhibit fms-like tyrosine kinase 3 (FLT3), Janus kinase (JAK), and Lck, suggesting the multidimensional therapeutic benefit of fostamatinib [122,123]. In mouse models of ITP and AIHA, fostamatinib was able to prevent the development of thrombocytopenia and anemia, respectively [124]. In two phase 3 randomized controlled clinical trials for ITP, FIT-1 and FIT-2 [125,126], fostamatinib demonstrated clinical benefits, with a statistically significant stable platelet response of 18% (defined as a platelet count of >50 × 10^9^/L, without rescue, for at least 4 of 6 weeks during treatment weeks 14–24) and an overall response (defined as any platelet count >50 × 10^9^/L within the first 12 weeks) of 43% vs. 14% in the placebo arm. Median time to response was 15 days, and 83% responded within 8 weeks. Responses were durable, with a median response duration of >28 months [126]. In April 2018, Fostamatinib was approved by the Food and Drug Administration (FDA) for the second-line treatment of chronic ITP.

A phase 2, multicenter, open-label SOAR study evaluated the response to fostamatinib in adult patients with warm AIHA who failed at least one prior therapy. The primary endpoint for hemoglobin of more than 10 g/dL, with an increment in hemoglobin of more than 2 g/dL at week 24, was achieved by 44% (11/24) of patients, with an additional responder at week 30. The hemoglobin response was rapid and durable. Responders were most often female, had secondary AIHA, and had fewer prior therapies for AIHA as compared to non-responders. AEs were mostly mild to moderate [127]. A phase 3 randomized, double-blind, global clinical trial investigating the safety and efficacy of fostamatinib in patients with warm AIHA is close to completion [128]. An extension study evaluating the long-term safety of fostamatinib is also ongoing [129].

#### 5.2.2. FcRn Inhibition

The neonatal Fc receptor for IgG (FcRn) is comprised of β2-microglobulin and a membrane-anchored α-chain related to the α-chain of a class I major histocompatibility complex [130]. It was initially identified as the receptor responsible for the transfer of passive humoral immunity from the mother to the fetus and newborn. FcRn also functions to protect IgG and albumin from catabolism, which accounts for their prolonged half-life compared to other immunoglobulins and liver-synthesized proteins, and plays a role in their transcellular transport [131]. FcRn is expressed in various tissues in adults, with the vascular endothelium as well as antigen-presenting hematopoietic cells accounting for most of the IgG recycling. IgG is internalized by FcRn-expressing cells through non-specific fluid-phase pinocytosis. FcRn binds tightly to the Fc portion of IgG at acidic but not physiological pH. Upon the acidification of the endosome, FcRn binds IgG, and the IgG–FcRn complex is recycled back into circulation, thus extending its serum half-life, while unbound IgG is destined for lysosomal degradation [132]. The FcRn-mediated IgG half-life extension is of benefit for antibody responses against pathogens; it is also responsible for the prolonged serum half-life of the pathogenic IgG autoantibodies, which promotes tissue damage in autoimmune diseases [133].

The removal or lowering of the circulating IgG is the main strategy in the treatment of IgG-mediated autoimmune disease. Treatments such as plasmapheresis, immunoadsorption, and immunosuppressant therapies are all the current modalities used to lower or eliminate the circulating IgG. IVIG is used widely in autoimmune diseases such as autoimmune thrombocytopenia purpura, Guillain–Barre syndrome, and Kawasaki disease. A proposed mechanism of these modalities is the oversaturation of FcRn, which results in the accelerated clearance of pathogenic antibodies [134]. As such, the blockade of the FcRn function diminishes the IgG circulatory half-life and has the potential to ameliorate antibody-mediated disorders. Several strategies have been used to inhibit FcRn, including antibodies blocking the IgG binding site, FcRn inhibitory peptides and small proteins, and the modification of its Fc regions to bind at neutral and acidic pH [135]. The pharmacokinetic modulation of ph-independent binding enhances FcRn inhibition. Several FcRn inhibitors are being studied in antibody mediated diseases, including wAIHA.

ALXN1830, also known as orilanolimab or SYNT001, is a humanized IgG4 antibody that blocks the binding of IgG and IgG immune complexes (ICs) to the FcRn, thus reducing the circulating IgG and IgG ICs [136]. ALXN1830 showed promising clinical efficacy in active, refractory pemphigus [137], and patients experienced a rapid improvement in symptoms within 14 days of the first dose, with acceptable adverse effects [138]. Trials for wAIHA were initiated but subsequently withdrawn [139,140,141].

Efgartigimod is a human IgG1 antibody Fc fragment which has been shown in a short, randomized phase 2 trial to induce the rapid reduction of IgG levels, which was associated with clinically relevant increases in platelet counts compared to placebo [142]. It is currently being investigated in a phase 3 study on ITP [143].

M281, also known as nipocalimab, is a high-affinity, fully human, glycosylated, effectorless monoclonal IgG1 anti-FcRn antibody that binds with picomolar affinity to FcRn to allow the occupancy of FcRn throughout the recycling pathway [144]. In an animal model, M281 has already demonstrated efficacy in the rapid clearance of IgG and IgG autoantibodies in immune cytopenia [145]. In a phase 2 multicenter, randomized, double-blind, placebo-controlled study on subjects with generalized myasthenia gravis, treatment with nipocalimab was associated with rapid and durable total IgG and pathogenic IgG reduction, which was correlated with a meaningful clinical response and was well-tolerated [146]. A phase 2/3 multicenter, randomized, double-blind, placebo-controlled clinical trial evaluating the efficacy and safety of M281 in warm AIHA [147] is currently enrolling patients [148]. In a preclinical model, M281 was able to block the transplacental transfer of alloimmune pathogenic IgG antibody [149], supporting the development of M281 for the treatment of alloimmune IgG-mediated fetal and neonatal diseases. The UNITY trial is a multicenter, open-labeled, phase 2 clinical study evaluating the safety of M281 in mothers and neonate/infants who are at high risk of early-onset severe hemolytic disease of the fetus and newborn (EOS-HDFN) [150].

RVT-1401 (Batoclimab) is a human recombinant anti-FcRn monoclonal IgG1 antibody that was developed for intravenous or subcutaneous administration [151], and which has demonstrated a dose-dependent reduction in the IgG level of healthy participants [152]. Currently, there are ongoing clinical trials for myasthenia gravis [151] and thyroid-associated ophthalmopathy [153]. ASCEND-WAIHA, a phase 2, non-randomized, open-label study evaluating the safety, tolerability, and efficacy of RVT-1401 in refractory warm AIHA was initiated but terminated by the sponsor in December 2021 [154].

### 5.3. Plasma Cell-Directed Therapy

B-cell-directed therapy with rituximab has been highly effective in relapsed and refractory cases of AIHA. For patients who do not respond to rituximab, the persistence of long-lived CD20 negative antibody secreting autoreactive plasma cells in the spleen has been proposed as a possible mechanism of resistance. In addition, B-cell depletion itself may promote the differentiation between short-lived and long-lived autoreactive plasma cells contributing to the resistance to therapy [155]. Using single-cell RNA sequencing analysis in patients with ITP who initially responded to rituximab and subsequently relapsed at the time of B-cell reconstitution, Cricks et al. [156] demonstrated the reactivation of a population of quiescent, autoreactive, rituximab-resistant splenic memory B cells, which are characterized by the down-regulation of B-cell-specific factors and the expression of pro-survival genes. Plasma cell inhibition offers the potential for antibody depletion therapy in autoimmune diseases such as AIHA alone or in combination with other agents.

#### 5.3.1. Proteasome Inhibition

Proteasome is an ATP-dependent, multicatalytic enzyme complex located in the cytoplasm and nucleus, which are responsible for the degradation of most intracellular proteins [157]. Proteins that are no longer required are tagged with ubiquitin, which then directs them to the proteasome where they are subsequently degraded [158]. Proteins degraded by the proteasome include the regulators of cell-cycle progression and apoptosis such as cyclins, caspases, B-cell lymphoma2, and the NF-kappa B endogenous inhibitor I-κB [159]. NF-kappa B upregulates transcription of genes involved in cell survival, adhesion, and cytokine signaling.

Bortezomib is a dipeptide boronic acid analog that reversibly inhibits the activity of the 20S subunit of the proteasome [158]. Proteosome inhibition leads to the accumulation of the important regulatory intracellular proteins, including I-κB, leading to decreased NF-kappa B activity and promoting apoptosis [160]. In preclinical studies, bortezomib had an immunomodulatory effect by decreasing the number of CD4+ T cells and reducing their production of TH1 cytokines, the impairment of B-cell function, antigen presentation, and antibody secretion by inducing the apoptosis of antibody-secreting plasma cells and memory B cells [161]. Given the multiple effects on the immune system, bortezomib is a promising agent in the treatment of autoimmune diseases.

Bortezomib has been reported to be effective in the setting of hematopoietic and solid organ transplant-associated AIHA in a number of case reports [162,163,164,165,166,167]. In retrospective case series and case reports on relapsed refractory wAIHA, bortezomib in combination with dexamethasone had an overall response rate of 85%, with five CR (35%) and seven PR (50.0%) [168,169,170] (Table 2). In a GIMEMA study—a phase 2 prospective, open-label, multicenter, clinical trial which evaluated the efficacy of a single course of bortezomib (1.3 mg/m^2^ on day 1, 4, 8, 11) in CAD—among the 19 heavily pretreated patients, the overall response rate was 31.6%, with 66.7% maintaining response after a median follow-up of 16 months, with low rates of toxicity [54] (Table 2).

Bortezomib has also been combined with other agents to treat AIHA, in the first-line setting with the hope of minimizing steroid exposure [171] as well as in the refractory setting in case reports and retrospective studies [172,173,174], most commonly in combination with rituximab. A combination of rituximab and bortezomib would synergically increase plasma-cell and B-cell depletion [174]. A clinical trial evaluating the efficacy of a single dose of rituximab combined with a short course of bortezomib in relapsed refractory AIHA is currently recruiting patients [175].

#### 5.3.2. CD38 Monoclonal Antibody

CD38 is a transmembrane glycoprotein with ectoenzymatic activity as well as receptor and adhesion molecule functions. It is strongly expressed in plasma cells, with weak expression in other lineages, including lymphoid, myeloid, and non-hematopoietic cells [176]. CD38 is involved in the series of reactions which contribute to nicotinamide adenine dinucleotide (NAD+) and nicotinamide adenine dinucleotide phosphate (NADP) metabolism, intracellular calcium mobilization, and the transduction of activation and proliferation signals [177]. In the context of multiple myeloma, a high level of expression of CD38 in plasma cells makes it a promising target for monoclonal antibody-based immunotherapy. CD38 antibodies kill myeloma cells via antibody-dependent cell-mediated cytotoxicity, complement-dependent cytotoxicity, antibody-dependent cell-mediated phagocytosis as well as the direct apoptosis of CD38+ MM cells via FcγR-mediated crosslinking [178,179]. Anti-CD38 therapy is a potential target for the treatment of systemic autoimmune disease by depleting long-lived antibody-producing plasma cells [180].

Daratumumab is a first-in-class, fully human IgG1-kappa monoclonal antibody targeting CD38 that is approved for the treatment of multiple myeloma [181]. In addition, daratumumab has been reported to be effective in post allogeneic bone marrow transplantation-related AIHA and Evan’s syndrome in several case reports [182,183,184,185]. Daratumumab is effective in both warm and cold primary AIHA based on current case reports and one retrospective study [186,187,188] (Table 2). Time to response was generally very short, with the majority of responses seen within 2 weeks. A majority of the patients who responded to daratumumab were heavily pretreated and were refractory to rituximab. The proposed efficacy is secondary to the depletion of long-living autoantibody-producing CD20 negative plasma cells [188]. Relapses were seen after therapy, suggesting the possible reemergence of CD20-positive CD38-negative memory B cells, prompting the consideration of dual CD20/CD38 inhibition in refractory cases. A phase I clinical trial evaluating the safety of daratumumab for the treatment of refractory AIHA is currently in progress [189].

Isatuximab is a novel immunoglobulin G1 kappa anti-CD38 monoclonal antibody that binds selectively to a specific epitope on CD38. It induces the internalization of CD38 from the myeloma cell surface, sensitizes myeloma cells to bortezomib, triggers antibody-dependent cellular cytotoxicity, antibody-dependent cellular phagocytosis, and complement-dependent cytotoxicity. In addition, it depletes CD38 B lymphocyte precursors and NK cells [190,191]. In March 2021 isatuximab was approved by the FDA for relapsed and refractory multiple myeloma [192]. Isatuximab is currently being studied in a multicenter, open-label, phase 1b/2 study on warm AIHA [193].

**Table 2 pharmaceutics-14-01035-t002:** Plasma cell-directed therapy.

Medication	Type of AIHA	Type of Study	Concurrent Disease	Regimen	Efficacy/Response Rate	Reference
Bortezomib	Mixed	Case report	Post allo-SCT	1.3 mg/m^2^ day 1, 4, 8, 11	Transfusion independent	[162]
	N/A	Case report	Post solid organ transplant	1.3 mg/m^2^ day 1	Transfusion independent	[163]
	N/A	Case report	Post allo-SCT	1.3 mg/m^2^ day 1, 8, 15, 22	Transfusion independent	[164]
	N/A	Case report	Post allo-SCT	1.3 mg/m^2^ day 1, 4, 8, 11 for 2 cycles	Transfusion independent	[165]
	Warm	Case report	Post solid organ transplant	1.3 mg/m^2^ day 1, 4, 8, 11 monthly	Transfusion independent	[166]
	Mixed	Case report	Post solid organ transplant	1.3 mg/m^2^ D 1, 4, 8, 11	CR	[167]
	N/A	Case report	SLE	1.3 mg/m^2^ D 1, 4, 8, cyclophosphamide, steroids	CR	[172]
	Warm	Retrospective, N = 7	N/A	1.3 mg/m^2^ D 1, 8, 15, 22, rituximab, steroids	ORR 85.71%	[171]
	Warm	Retrospective, N = 7	N/A	1.3 mg/m^2^ D 1, 4, 8, 11, rituximab	CR 71.4%	[173]
	Warm	Retrospective, N = 8	N/A	1.3 mg/m^2^ D 1, 4, 8, 11, steroids	ORR 75%	[168]
	Warm	Case report	N/A	1.3 mg/m^2^ D 1, rituximab	PR	[174]
	Warm	Case series, N = 2	N/A	1.3 mg/m^2^ D 1, 4, 8, 11, dexamethasone	PR 2/2	[169]
	Warm	Case series, N = 4	N/A	1.3 mg/m^2^ D 1, 4, 8, 11, dexamethasone	CR 1/4, PR 2/4	[169]
	Cold	Open-label, phase 2	42.9% B-LPD	1.3 mg/m^2^ D 1, 4, 8, 11	ORR 31.6%	[54]
Daratumumab	N/A	Case report	Post allo-SCT	16 mg/kg weekly in 4 doses	CR	[182]
	N/A	Retrospective, N = 3	Post allo-SCT	N/A	CR 67%	[183]
	N/A	Retrospective, N = 3	Post allo-SCT	N/A	CR 67% PR 33%	[184]
	N/A	Case report	Post allo-SCT	16 mg/kg weekly in 6 doses	CR	[185]
	Warm	Case report	N/A	16 mg/kg weekly	CR	[186]
	Warm	Retrospective, N = 4	50% post allo-SCT	16 mg/kg weekly	ORR 100%, CR 50%	[188]
	Cold	Case report	N/A	16 mg/kg weekly	Response	[187]

allo = allogeneic, B-LPD = B-cell lymphoproliferative disorder, CR = complete response, D = day, ORR = overall response rate, PR = partial response, SCT = stem cell transplant.

### 5.4. Complement Inhibition

Complement-mediated hemolysis is the key pathway responsible for red blood cell destruction in cold agglutinin disease, and it plays a role in up to 50% of patients with wAIHA [16,113]. Antigen–antibody complexes activate the classical complement pathway via recruitment of the complement component C1q. IgM is the most effective antibody isotype in activating a complement. IgG subclasses differ in their ability to activate complement, with IgG3 being the most efficient, followed by IgG1, IgG2, and IgG4. The activation of C1q leads to the activation of C1r and a serine protease C1s. C1s subsequently activates C4 and C2, generating the C3 convertase, which cleaves C3 to C3a and C3b. Intravascular hemolysis occurs when a large amount of complement is generated to form the membrane attack complex. However, the presence of complement regulatory proteins CD55 and CD59 on the surface of red blood cells could generally inhibit this process. The C3b-coated RBCs are sequestered by macrophages of the reticuloendothelial system, predominantly in the liver (extravascular hemolysis) [194]. Complement inhibitors demonstrated efficacy in cold agglutinin disease and wAIHA with evidence of complement involvement.

Eculizumab is a monoclonal antibody that targets the complement protein C5 and prevents membrane attack complex formation and the resultant intravascular hemolysis. In patients with paroxysmal nocturnal hemoglobinuria (PNH), a hemolytic anemia caused by an acquired deficiency of the terminal complement regulators CD55 and CD59, therapy with eculizumab was associated with a significant reduction of intravascular hemolysis, thrombotic events, and transfusion requirements, as well as the improvement of anemia, dyspnea, fatigue, and quality of life [195,196]. Eculizumab has been reported to be effective in reducing hemolysis and the transfusion requirement in patients with CAD as well as control hemolytic crisis in the setting of infection [197,198,199]. In an open-label, prospective phase 2 trial, eculizumab significantly reduced hemolysis in 7 out of 13 patients with CAD, and 8/9 transfusion-dependent patients achieved transfusion independence. The increase in hemoglobin (median, 0.8 g/dL) was modest, thus supporting the key role of extravascular hemolysis in CAD. Cold-induced circulatory symptoms such as acrocyanosis or Raynaud syndrome were not affected by eculizumab [18].

Pegcetacoplan (APL-2) is a pegylated cyclic peptide and a compstatin analog that targets complement C3, which is more proximal in the complement cascade, inhibiting both intravascular and extravascular hemolysis [200]. In a randomized, multicenter, open-label phase 3 clinical trial involving patients with PNH, pegcetacoplan therapy was associated with a significantly greater improvement in hemoglobin level, a reduction in the need for transfusion, control of hemolysis, and improvement in fatigue as compared to treatment with the C5 inhibitor eculizumab. The incidence of side effects was low, mainly injection-site irritation and mild diarrhea, and there were no meningococcal infections [201]. Pegcetacoplan was approved for the treatment of patients with PNH in May 2021 [202]. Pegcetacoplan is being evaluated in the ongoing PLAUDIT study, a Phase 2 open-label study involving patients with primary AIHA. In a preliminary report which included 13 patients with CAD, mean hemoglobin increased from 8.9 to 11.6 g/dL, with the normalization of hemolysis markers and the improvement in FACIT scores. Mean hemoglobin increases were more modest, from 9.2 to 10.8 g/dL among eight patients with wAIHA and monospecific DAT positive for C3d [203,204,205].

Wouters et al. demonstrated a dose-dependent inhibition of C3 deposition in the sera of patients with IgM-mediated AIHA by the C1 inhibitor (C1-INH) concentrate in vitro. In a patient undergoing therapy for severe refractory warm IgM-antibody-mediated AIHA in the setting of an aggressive, diffuse large B-cell non-Hodgkin lymphoma, C1-INH administration resulted in the attenuation of complement deposition on RBCs and controlled hemolysis [206]. Subsequently, TNT003, a mouse monoclonal anti-C1s antibody that targets C1s serine protease activity, has been shown to block classical complement pathway activation and prevent C3d deposition and phagocytosis of red blood cells exposed to CAD patient plasma; it also completely inhibited hemolysis in vitro [16]. This led to the development of the humanized monoclonal antibody, sutimlimab (TNT009 or BIVV009).

Sutimlimab selectively targets the C1s, a serine protease responsible for the activation of the classical complement pathway. By selectively inhibiting the classical pathway upstream at C1s, sutimlimab does not inhibit the immune function of the lectin and alternative complement pathways with potential for less immunosuppression. In a phase 1b study, sutimlimab rapidly increased hemoglobin levels in 7 out of 10 patients with cold agglutinin disease and was associated with the normalization of the hemolysis markers. Hemolysis recurred upon discontinuation of the drug, and a rechallenge with sutimlimab in a named-patient program recapped the control of hemolytic anemia and transfusion independence [207,208].

In a multicenter, open-label, single-group study involving 24 patients with cold agglutinin disease and with a recent history of transfusion, sutimlimab administered at a fixed, biweekly dose of 6.5 or 7.5 g in those weighing <75 or ≥75 kg, respectively, resulted in the rapid normalization of hemolysis. A mean hemoglobin level of more than 11 g per deciliter was maintained in patients from week 3 through the end of the study period, with 17 patients (71%) achieving transfusion independence after week 5. Increased hemoglobin levels, reduced bilirubin levels, and reduced fatigue coincided with the inhibition of the classical complement pathway and were associated with a clinically meaningful reduction in fatigue. No meningococcal infections or other serious adverse events related to the study drug was noted [51]. A randomized, double-blind, placebo-controlled, randomized phase 3 trial of sutimlimab involving non-transfused patients with CAD has completed accrual [209]. Preliminary results were presented at European Hematology Association Congress in June 2021, with 73% (*n* = 16) of patients treated with sutimlimab meeting the primary composite endpoint of improvement in hemoglobin by ≥1.5 g/dL from baseline; transfusion independence after week 5; and avoidance of other CAD-related treatments compared to 15% (*n* = 3) in the placebo group (Odds Ratio = 15.9, *p* < 0.001). Responses were rapid, with a mean increase from baseline of ≥1 g/dL by Week 1 and ≥2 g/dL by Week 3; this was sustained throughout the duration of treatment and is associated with the improvement of hemolysis markers and improvement of fatigue. The improvements coincided with near-complete classical pathway inhibition and C4 normalization [210]. In February 2022, sutimlimab became the first FDA-approved therapy for the treatment of cold agglutinin disease. A phase 1b study accessing the safety and tolerability study of the anti-C1s antibody BIVV020 has recently been completed [211].

ANX005 is a humanized monoclonal antibody (ANX005) that binds with high-affinity to C1q, the initiating molecule of the complement cascade. In a preclinical study, ANX005 reduced hemolysis and C3 and C4 complement deposition on human RBCs that were pre-sensitized with sera from CAD subjects [212]. In the sera of patients suffering from wAIHA with a DAT positive for IgG and C3, Teigler et al. demonstrated the ability of autoantibodies to induce C4 complement deposition on the surface of donor RBCs. The addition of ANX005 fully blocked the deposition of C4 on RBCs by wAIHA sera, indicating the dependence of the classical complement pathway and the therapeutic potential of blocking classical complement pathway activity in wAIHA [213]. The therapeutic potential of blocking classical complement pathway activity with an anti-C1q inhibitor in wAIHA patients who show evidence of classical complement pathway activity is currently being evaluated in an ongoing Phase 2 interventional trial [214].

The limitations of complement inhibition include the need for ongoing therapy to control hemolysis. In addition, ischemic symptoms such as Raynaud’s phenomenon, which are seen in majority of patients with CAD, are not complement-mediated and will not be alleviated. Complement inhibitors would be particularly helpful in severely anemic patients within the setting of acute exacerbations of hemolysis, in patients with CAD for whom chemoimmunotherapy has failed or is contraindicated, or as a bridge to B-cell-directed treatment.

## 6. Conclusions

AIHA is a heterogeneous group of disorders characterized by antibody mediated red blood cell destruction. Warm and cold autoimmune hemolytic anemias differ in pathophysiology and treatment options. The treatment of warm autoimmune hemolytic anemia involves corticosteroids, rituximab, splenectomy, and other immunosuppressive agents. Cold agglutinin disease therapies are directed at the eradication of the pathogenic clone with rituximab, with the addition of bendamustine in selected cases and the inhibition of hemolysis with the recently approved complement inhibitor sutimlimab. Relapses are common, and there is an unmet need for subsequent therapies. Emerging therapies for autoimmune hemolytic anemia target various steps in the pathogenesis of AIHA and include B-cell-directed therapies with BTK inhibitors, PI3K inhibitors, mTOR inhibitors, phagocytosis inhibition with the Syk inhibitor, FcRn inhibitors, plasma cell-directed therapies with proteasome inhibitors and CD38 monoclonal antibodies, and complement inhibition. The existing data on novel therapies in heavily pretreated AIHA patients are scarce but promising, with an acceptable toxicity profile. Further prospective studies are warranted in order to evaluate efficacy and safety of novel therapies for autoimmune hemolytic anemia.

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
