# Peer review of "Development of New Drugs for Autoimmune Hemolytic Anemia"

_pharmaceutics, 2022, doi:10.3390/pharmaceutics14051035_

Round 1

Reviewer 1 Report

 Xiao and Murakhovskaya have submitted a focused review on treatment of autoimmune hemolytic anemia (AIHA), emphasizing the development of new drugs. The AIHA classification on which the review is based is up-to-date, as is the literature cited. The unmet needs with existing therapies are clearly explained, and the discussion of new and emerging drugs is comprehensive and complete. The paper is well-written and interesting. This is an excellent article and I have only a couple of minor comments.

Minor comments:

  1. Page 4, Treatment of CAD: Considering that the percentage of CAD patients with compensated hemolysis or mild anemia is 36% (12+24%) according to one of the authors’ references, I would delete the word ‘small’ from line 139.
  2. A few of the references are incomplete (#127, #187 and, possibly, others). Please check references for completeness.
  3. A prospective study of fostamatinib in wAIHA, cited by the authors as a preliminary reference, has recently been published as a full article (https://doi.org/10.1002/ajh.26508). I suggest you update this reference.

Author Response

We thank referee 1 for the kind remarks. We have addressed all the comments you mentioned

  1. Page 4, Treatment of CAD: Considering that the percentage of CAD patients with compensated hemolysis or mild anemia is 36% (12+24%) according to one of the authors’ references, I would delete the word ‘small’ from line 139.

The word "small" was removed from new line 163

2. References were updated including reference #127, #187 and checked for completeness.

3. A prospective study of fostamatinib in wAIHA, was updated (reference 131)

Reviewer 2 Report

This review is very complete and exhaustive description of the fundamental aspects to be considered for the treatment of autoimmune hemolytic anemias (AHA) due to both warm and cold autoantibodies.

In its current format, this review provides basic and useful information for the general hematologist, but also a detailed description of the most recent drugs for the treatment of patients with difficult diagnosis AHA or that does not respond to the conventional drugs or splenectomy. There are also interesting indications for the follow up of patients with autoimmune hemolytic anemia.

Author Response

We really appreciate referee 2 for the kind remarks.

Reviewer 3 Report

This is an interesting review addressing new treatments for Autoimmune hemolytic anemia (AIHA).

They discuss traditional first-line immunosuppressive therapy including corticosteroids and rituximab is associated with adverse effects, treatment failures and relapses, as well as subsequent lines of therapy that are associated with higher rates of toxicity, and some patients remain refractory to currently available treatments.

Finally, they discuss novel therapies that have become promising for this vulnerable population. Since they discuss the mechanism of action, existing data and ongoing clinical trials of current novel therapies for AIHA, they should provide more detail on the mechanism of action of classical therapies.

Subsequent therapies: it would be of interest to report the mechanism of action of those agents, especially azathioprine which is frequently used as maintenance treatment, as reported and discussed in a review on its use in another autoimmune chronic disease (Diagnosis and therapy of autoimmune hepatitis. Mini Rev Med Chem. 2009 Jun;9(7):847-60.).

Author Response

We thank referee 3 for the important remarks. We have added the mechanism of corticosteroids in lines 102-103 and the mechanism of splenectomy lines 138-142. We have also added the mechanism of action of these conventional immunosuppressive agents including azathioprine in subsequent therapies part lines 145-154

Reviewer 4 Report

This is a review of the new therapeutic options available for AIHA. In recent years, many new therapies have been developed for this pathology and many reviews have recently been published on the same topic. This review is detailed, sufficiently complete and well written but it does not add much to this panorama. 

Author Response

Thank you referee 4 for the remarks. We also acknowledge that many reviews have been published in this area recently including the most updated how I treat series in 2021. We focus on the novel treatment for both warm and cold AIHA and especially included the ongoing trials for these novel agents which is unique in this review